# Preclinical evaluation of a protracted GLP-1/ glucagon receptor co-agonist: Translational difficulties and pitfalls

Lotte Simonsen[1], Jesper Lau[2], Thomas Kruse[2], Tingqing Guo[3], Jim McGuire[4¤], Jacob Fuglsbjerg Jeppesen[5], Kristoffer Niss[6], Per Sauerberg[7], Kirsten Raun[1], Charlotta Dornonville de la Cour[5]*

**1** Global Obesity & Liver Disease Research, Novo Nordisk A/S, Måløv, Denmark, **2** Research Chemistry, Novo Nordisk A/S, Måløv, Denmark, **3** Discovery Biology, Novo Nordisk Research Centre, Beijing, China, **4** Incretin Biology, Novo Nordisk A/S, Måløv, Denmark, **5** Global Diabetes, Cardio- & Renal Research, Novo Nordisk A/S, Måløv, Denmark, **6** Bioinformatics & Data Mining, Novo Nordisk A/S, Måløv, Denmark, **7** Project and Alliance Management, Novo Nordisk A/S, Måløv, Denmark

¤ Current address: Translational Research, Catalyst Biosciences, San Francisco, CA, United States of America

* cddc@novonordisk.com

**Data Availability Statement:** All relevant data are within the paper and its Supporting Information files.

**Funding:** The work was funded by Novo Nordisk.

## Abstract

During recent years combining GLP-1 and glucagon receptor agonism with the purpose of achieving superior weight loss and metabolic control compared to GLP-1 alone has received much attention. The superior efficacy has been shown by several in preclinical models but has been difficult to reproduce in humans. In this paper, we present the pre-clinical evaluation of NN1177, a long-acting GLP-1/glucagon receptor co-agonist previously tested in clinical trials. To further investigate the contribution from the respective receptors, two other co-agonists (NN1151, NN1359) with different GLP-1-to-glucagon receptor ratios were evaluated in parallel. In the process of characterizing NN1177, species differences and pitfalls in traditional pre-clinical evaluation methods were identified, highlighting the translational challenges in predicting the optimal receptor balance in humans. In diet-induced obese (DIO) mice, NN1177 induced a dose-dependent body weight loss, primarily due to loss of fat mass, and improvement in glucose tolerance. In DIO rats, NN1177 induced a comparable total body weight reduction, which was in contrast mainly caused by loss of lean mass, and glucose tolerance was impaired. Furthermore, despite long half-lives of the three co-agonists, glucose control during steady state was seen to depend on compound exposure at time of evaluation. When evaluated at higher compound exposure, glucose tolerance was similarly improved for all three co-agonists, independent of receptor balance. However, at lower compound exposure, glucose tolerance was gradually impaired with higher glucagon receptor preference. In addition, glucose tolerance was found to depend on study duration where the effect of glucagon on glucose control became more evident with time. To conclude, the pharmacodynamic effects at a given GLP-1-to-glucagon ratio differs between species, depends on compound exposure and study length, complicating the identification of an optimally balanced clinical candidate. The present findings could partly explain the low number of clinical successes for this dual agonism.

**Competing interests:** All authors are or were full time employees at Novo Nordisk A/S and hold a minor share portion as part of their employment. Novo Nordisk A/S manufactures and markets pharmaceuticals related to diabetes, obesity and other chronic diseases and has intellectual property rights for several inventions related to these diseases. This does not alter our adherence to PLOS ONE policies on sharing data and materials.

## Introduction

Obesity is a disease associated with serious metabolic health consequences including type 2 diabetes, dyslipidaemia, cardiovascular disease and cancer. The prevalence of obesity is increasing both in developed and developing countries [1, 2]. Despite years of extensive research in the field, few pharmacological therapies exist, and the weight loss obtained is still modest compared with the weight loss obtained by bariatric surgery [3].

Unimolecular receptor co-agonism is an approach to potentially obtain a superior weight loss by targeting several mechanisms of action simultaneously using a single molecule. One combination that has received much attention is the combination of GLP-1 and glucagon receptor agonism, as first described in 2009 [4, 5] and thereafter intensively investigated by several (for a recent review see [6]). GLP-1 and glucagon are products of the proglucagon gene arising from tissue-specific post-translational processing and share a large degree of sequence homology [7]. Successful co-agonism can therefore be obtained by modifying a relatively small number of amino acids using either glucagon or GLP-1 as starting backbone.

GLP-1, produced and secreted by intestinal L-cells in response to nutrients, improves glucose homeostasis [7], while simultaneously lowering body weight by reducing energy intake [8]. Several long-acting GLP-1 receptor agonists have been clinically approved for the treatment of type 2 diabetes [9] and more recently also for the treatment of obesity (Saxenda® and Wegovy®).

Glucagon, produced primarily in pancreatic α-cells, is traditionally viewed as a hormone counter-regulating insulin, preventing hypoglycemia by increasing hepatic glucose production [10]. It has also been acknowledged for its ability to reduce food intake and increase energy expenditure, lipolysis, fatty acid oxidation and ketogenesis [11]. In addition, evidence exists to support a central role for glucagon in regulation of amino acid metabolism [12]. Several of these metabolic effects make glucagon attractive as an anti-obesity agent, but the applicability, especially in type 2 diabetic subjects, is complicated by the inherent risk of inducing hyperglycemia.

The rationale for combining GLP-1 and glucagon is therefore to utilize the anti-obesity properties of glucagon with the appetite-regulating and anti-diabetic properties of GLP-1. If combined in the optimal balance, the elicited receptor responses should yield a pronounced body weight reduction while maintaining optimal glucose control.

In humans, short-term and acute studies with the endogenous co-agonist oxyntomodulin, as well as combinations of native GLP-1 and glucagon support the rationale for combining the mechanism of action of the two peptides [13–15]. In addition, several longer-term studies performed in mice and non-human primates succeeded to balance the two components resulting in superior metabolic benefits compared with single receptor agonism alone [5, 16–19]. In an extensive study by Day *et al.*, several co-agonists with different GLP-1-to-glucagon ratios were investigated in mice and the optimal balance was suggested to be a co-agonist with equal potency on both receptors [20]. Based on the promising preclinical results several pharmaceutical companies have taken co-agonists, both balanced and with varying degree of GLP-1 receptor selectivity into clinic [6]. However, the superior efficacy of the dual agonism found in mice have been difficult to reproduce in humans, which has resulted in the termination of several clinical co-agonist projects [6].

In this paper we describe the pre-clinical characterization of NN1177, a long-acting GLP-1/ glucagon receptor co-agonist. In order to better interpret results and further explore the role of GLP-1 and glucagon receptor agonism, one GLP-1 reference compound, one co-agonist with higher GLP-1 receptor selectivity (NN1359) and one co-agonist with higher glucagon-receptor selectivity (NN1151) were evaluated in parallel. Our findings illustrate the difficulty

of identifying the optimal ratio even in preclinical models which further highlights the challenges in a clinical evaluation.

## Materials and methods

### Compounds

NN1151 and NN1177 (NNC9204-1177) are based on the human glucagon sequence while NN1359 is based on the human GLP-1 sequence. Several amino acid substitutions were made to the glucagon backbone to enhance the GLP-1 receptor agonism of NN1151 and NN1177. Of pivotal importance was changing the C-terminal carboxylic acid to a carboxamide. The truncation of the C-terminus in combination with a carboxamide was important for obtaining co-agonism for the GLP-1-based sequence NN1359. A GLP-1 reference compound with preferential GLP-1 receptor potency was generated based on the glucagon sequence in order to be structurally as close to NN1177 as possible. The peptide sequences are depicted in Fig 1. In addition to changes necessary for dual receptor activity, several amino acid substitutions were introduced to ensure long-term formulation stability. A longer plasma half-life was secured by the attachment of a C18 fatty diacid, in line with the once-weekly GLP-1 receptor agonist semaglutide [21]. However, in contrast to semaglutide side-chain, the linker between the peptide backbone and the fatty acid contains several negatively charged residues. Incorporation of such residues were generally shown to improve physical stability of glucagon-based co-agonists when evaluated in a thioflavin T-based fibrillation assay [22]. The compound synthesis is described in detail in S1 File. When used for *in vitro* assays the compounds were dissolved in 80% DMSO/20% water prior to further dilutions in assay buffers. The final concentration of DMSO was kept at a maximum of 0.3% in all assays. When used for *in vivo* assays, the compounds were dissolved in 50 mM sodium phosphate, 70 mM sodium chloride and 0.05% polysorbate 80; pH 7.4. This formulation was also used as vehicle.

### *In vitro* assays

The cell lines used for receptor binding affinity and potency assays were made by sequential, stable transfection of BHK (wild type) cells with the pGL4.29[luc2P/CRE/Hygro] plasmid (Promega, Madison, WI) followed by a second transfection with a plasmid encoding either the glucagon receptor or the GLP-1 receptor from the relevant species. The binding assay was performed on cell membranes in the presence of 1% (v/v) species-specific heparin-treated pooled plasma (Bioreclamation IVT, Westbury, NY) diluted in assay buffer (50 mM HEPES supplemented with 5 mM EGTA, 5 mM MgCl$_2$, and 0.005% (v/v) Tween 20, pH7.4). The receptor

**Fig 1. Depiction of compound sequences *vs* native human GLP-1 and glucagon.** The sequences of the GLP-1/glucagon receptor co-agonists (NN1177, NN1151, NN1359) and the GLP-1 reference. Red colour is used to denote amino acids arising from neither the human-glucagon or -GLP-1 sequence. X = 2-aminoisobutyric acid, B = 1-amino-1-cyclobutanecarboxylic acid, Z = 3-(1H-Imidazol-4-yl) propionic acid. The fatty acid containing side-chains are attached to the peptide sequences via the epsilon amine of lysine in positions marked with* (Sidechains used: NN1177; -γGlu-γGlu-Ser-Glu-Ser-γGlu-γGlu-C18 diacid, NN1151; -γGlu-γGlu-ADO-ADO-γGlu-γGlu-C18-diacid, GLP-1 ref.; -γGlu-γGlu-ADO-ADO-γGlu-γGlu-C18-diacid, NN1359; -ADO-ADO-γGlu-γGlu-C18-diacid. (For full structures see S1 Fig).

binding affinities of the compounds were measured in a scintillation proximity assay (SPA) bead-based (wheat germ agglutinin SPA beads (RPNQ0001), PerkinElmer, Waltham, MA) competitive binding assay using either [$^{125}$I]-glucagon or [$^{125}$I]-GLP-1(7–36)-NH$_2$ as tracers. Cells stably expressing an individual receptor of interest were grown at 5% $CO_2$, 37˚C in DMEM (supplemented with 10% FCS, 0.5 mg/ml G418 and 0.3 mg/ml hygromycin). Membrane preparations were made from cells harvested at approximately 80% confluence, washed and homogenized in 20 mM HEPES with 0.1 mM EDTA, pH 7.4. Protein concentration was determined, and the membranes were stored at -80˚C until use. The membrane binding assay was performed in a 96-well OptiPlate (PerkinElmer) in a total volume of 200 μl. Test compounds were dissolved in 80% DMSO and further diluted in assay buffer. 50 μl of diluted plasma was added to the assay plate wells followed by addition of test compounds (25 μl), cell membranes (50 μl, 0.06 mg/ml), SPA beads (20 mg/ml in assay buffer) and radioligand (60,000 cpm/well, corresponding to 0.06 nM; 2200 Ci/mmol). Each assay plate was incubated for 2 h at 30˚C, centrifuged at 1500 rpm for 10 minutes and counted in a TopCount NXT instrument (PerkinElmer).

Functional assays were performed on whole cells in the absence or presence of 100% species-specific heparin-treated pooled plasma (Bioreclamation IVT). Native GLP-1 and glucagon are rapidly degraded in presence of plasma and were therefore not measured in these assays. The cells were cultured with DMEM (supplemented with GlutaMAX, 10% FBS, 0.5 mg/ml G418 and 0.3 mg/ml hygromycin) at 5% $CO_2$ and 37˚C. At approximately 80–90% confluence, cells were washed with PBS and harvested with Versene (Life Technologies, Carlsbad, CA). The cells were counted and plated into 96-well microtiter plates coated with poly-D-Lys (Corning BioCoat) with 5,000 cells/well and thereafter incubated overnight in growth media. The next day the media was changed to assay buffer (DMEM without phenol red, supplemented with GlutaMAX, 10 mM HEPES, 1% ovalbumin and 0.1% Pluronic F-68) or to 100% plasma. In the absence of plasma, compounds were diluted in assay buffer. In assays tested with plasma, compounds were diluted in pure plasma. Buffer was removed from the plates and 50 μl of test compound dilutions was added to the plates followed by 50 μl of plasma or the same buffer used for compound dilution. The assay plates were incubated for 3 h in a 5% $CO_2$ incubator at 37˚C. The plates were then transferred to room temperature for 15 min and 100 μl Steadylite Plus reagent (PerkinElmer) was added to each well. The plates were shaken for 30 min at room temperature while protected from light after which luminescence was measured.

## Animals/Ethics

Animal studies were approved and conducted according to the permissions granted by the Danish Animal Experiment Inspectorate (permission numbers 2012-15-2934-00378, 2013-15-1934-00875) or by the local IACUC of Novo Nordisk Research Centre, Beijing, China. All animals were observed daily with special focus on the degree of body weight loss to ensure that no animals experienced a weight loss exceeding 20% relative to the body weight of a lean, age-matched control animal. All animals were euthanized with cervical dislocation under isofluran anaesthesia or with a combination of $CO_2$ and $O_2$ to ensure minimal suffering of the animals.

Male C57Bl/6J mice were purchased from Jackson Lab (Bar Harbour, ME) or Taconic, (Denmark). Male Sprague Dawley rats were purchased from Taconic (Denmark). Lean animals were fed regular chow diet (Altromin 1324, Brogaarden, Denmark). Obese animals were fed a high-fat diet, 45% kcal from fat (D12492, Research Diets, New Brunswick, NJ) from the age of 4–6 weeks in order to obtain a diet induced obese (DIO) phenotype. Animals were housed under temperature and humidity-controlled conditions in a 12:12 hr light:dark cycle

(lights on at 6 a.m.). Rats were housed three per cage during high fat feeding and were single housed five weeks prior to study. Mice were single housed, two per cage, with a partitioning wall where possible. All animals had *ad libitum* access to diet and water, except when fasted prior to experimental procedures. Animals were acclimatized to study conditions (energy expenditure cages, daily mock handling etc) at least one week prior to first dose. Animals were allocated into treatment groups to ensure comparable baseline body weight and body composition distribution between groups. At the time of study, animals were 20–30 weeks old with a body weight of 45–50 g (DIO mice) or approximately 600 g (DIO rats).

### *In vivo* experimental designs

**Compound exposure.** Compound exposure was measured by Luminescence Oxygen Channeling Immunoassay (LOCI). Donor beads were coated with streptavidin, while acceptor beads were conjugated with a monoclonal antibody specific for an epitope within the relevant molecule (NN1177 and NN1151: epitope in the N-terminal region; NN1359: epitope in the C-terminal region; GLP-1 ref: specific for the γ-Glu residues in the linker). The secondary monoclonal antibody, recognizing a different epitope on each of the molecules (NN1177: specific for the γ-Glu residues in the linker; NN1151: epitope in the C-terminal region; NN1359: specific for the N-terminal epitope; GLP-1 ref: epitope in the mid/N-terminal region of the molecule), was biotinylated. The three reactants were combined with the analyte and formed a two-sited immuno-complex. Illumination of the complex released singlet oxygen atoms from the donor beads. They were channeled into the acceptor beads and triggered a chemiluminescence response which was measured in an EnVision plate reader (PerkinElmer). The amount of light was proportional to the concentration of the analyte. As calibrators, a dilution line of the relevant analyte in rat/mouse plasma was included.

**Acute effect on glucose tolerance in DIO mice.** DIO mice (n = 8–9) were administered one or two subcutaneous (s.c.) injections of vehicle, 5 nmol/kg of the GLP-1 reference compound or 3 or 5 nmol/kg of NN1177. All animals received a single s.c. injection of compound at 4 pm, 21 h prior to administration of glucose (for all treated animals to have equally affected food intake in the dark period prior to the intraperitoneal glucose tolerance test (IPGTT)). Animals to be evaluated at the time of high plasma concentrations ($C_{max}$) received an additional injection of compound at 7 am, 6 h prior to glucose injection.

Animals were fasted for 5–6 h prior to intraperitoneal (i.p.) injection of 2 g/kg glucose. Blood glucose was analysed on whole blood from the tail vein of conscious animals using a Biosen S-line glucose analyzer (EKF-diagonistic Gmbh, Barleben, Germany) by the glucose oxidase method, at basal and at t = 15, 30, 60, 90, 180 min post glucose administration.

**Subchronic effect on body weight and glucose tolerance in DIO mice.** DIO mice received s.c. injections once daily for 4–5 weeks. Animals were dosed at different times of day to allow for assessment of glucose tolerance and other metabolic parameters at same time of day but at different time points in relation to dosing. Animals were assessed at a time of high compound exposure, 5 h post dosing ($C_{max}$) and at a time of lowest compound exposure, 24 h post-dosing ($C_{min}$). Body composition was assessed in conscious animals by Magnetic Resonance (MR) scanning prior to initiation of treatment and at the end of the treatment period. Steady-state plasma parameters were assessed in non-fasted animals. Additional DIO mice (n = 4) were included for verification of 24-h peptide exposure in steady state. The mice received s.c. doses daily for 2 weeks (13 days) with 3 nmol/kg NN1177; 5 nmol/kg NN1151 or 5 nmol/kg NN1359, where after plasma sampled after t = 5 h (NN1177) or 6 h (NN1151, NN1359), 10 h (all co-agonists) and 24 h (all co-agonists) post-dosing was analysed for exposure.

*Analogues were tested for subchronic effect in two different set-ups*. NN1177 vs GLP-1 reference: DIO mice (n = 10) received daily injections of vehicle, 5 nmol/kg of the GLP-1 reference or 3, 4, or 5 nmol/kg of NN1177 for 5 weeks. Different groups of NN1177-treated animals were evaluated at $C_{max}$ (dosed daily at 8 am and evaluated at 1 pm, 5 h post-dosing) and $C_{min}$ (dosed daily at 1 pm and evaluated at 1 pm, 24 h post-dosing). The $C_{max}$ and $C_{min}$ animals were included in separate treatment groups and evaluated head-to-head for glucose tolerance and additional metabolic parameters at the end of the treatment period. GLP-1 reference-treated animals were dosed daily at 8 am for evaluation at $C_{max}$. IPGTT was performed as described above, but additionally blood was sampled from vena facialis of conscious animals at t = 210 min relative to the administration of glucose for exposure analysis.

*NN1151 and NN1359 vs GLP-1 reference*. DIO mice (n = 9) received daily administrations of vehicle, 5 nmol/kg of GLP-1 reference, 1, 3, 5, 10, or 30 nmol/kg of NN1151 or 1, 3, 5, 10, or 30 nmol/kg of NN1359 for 4 weeks. The same groups of animals were evaluated at $C_{max}$ (dosed daily at 8 am and evaluated at 1 pm, 5 h post-dosing) after 3 weeks of dosing and one week later at $C_{min}$ (dosed daily at 1 pm and evaluated at 1 pm, 24 h post-dosing). IPGTT was performed as described above.

**Subchronic effect on energy expenditure in DIO mice.** DIO mice (n = 8) were treated once daily s.c. in the morning for eleven days with vehicle or 5 nmol/kg NN1177. Forty-eight hours prior to first dose and during the study, the metabolic phenotype was determined using the PhenoMaster system (TSE Systems GmbH, Bad Homburg, Germany). During this period, air flow, temperature, oxygen- and carbon dioxide content, oxygen uptake ($V_{O2}$) and carbon dioxide production ($V_{CO2}$) were simultaneously measured using standard indirect calorimetry analysis. The respiratory exchange rate (RER) and energy expenditure (EE) were calculated automatically from $V_{O2}$ and $V_{CO2}$. Furthermore, voluntary activity in x-y dimensions was determined. All parameters were measured simultaneously for one min per cage in two cages at a time, resulting in a time resolution of data points of 9 min.

**Subchronic effect on body weight and glucose tolerance in DIO rats.** DIO rats (n = 10) received once daily s.c. injections with either vehicle, 3 nmol/kg GLP-1 reference, 3 nmol/kg of NN1177 or a combination of 3 nmol/kg GLP-1 reference and 3 nmol/kg NN1177 for 3 weeks. Oral glucose tolerance was evaluated on study day 15, 5 h post-dosing (at $C_{max}$). Animals were fasted for 5 h prior to oral administration of glucose (3 g/kg). Blood glucose was analysed on whole blood from the sublingual plexus at baseline and at t = 15, 30, 60, 90, 120 and 180 min post glucose administration. Conscious animals were MR-scanned prior to initiation of treatment and at the end of the treatment period.

## Data analysis, calculations and statistical analysis

For the receptor binding assays, $logIC_{50}$ values were calculated using non-linear regression analysis. The error estimation was reported as the 95% confidence interval. In the potency assays, $logEC_{50}$ values were calculated by non-linear regression analysis using a 4-parameter model with the Hill slope constrained to be <2. The calculations were performed in GraphPad Prism 6.4 (GraphPad, San Diego, CA) and error estimation was expressed as the 95% confidence interval. The *in vitro* GLP-1- to-glucagon receptor ratios were calculated on the $EC_{50}$ level after normalization to the endogenous ligands on each receptor, in absence of plasma, to account for slight differences in the assay systems and reported as the ratio of GLP-1R:GCGR (glucagon receptor) $EC_{50}$ values. Averages, error estimation, ratios and plasma shifts were all calculated on $logIC_{50}$ or $logEC_{50}$ values and transformed to the anti-log values for reporting purposes.

All *in vivo* data are presented as mean ±SEM. For visualization and analysis regarding body weight, body fat and food intake of NN1177-treated animals, $C_{max}$ and $C_{min}$ groups within each treatment and dose level were pooled as no statistical differences were observed between the two different dosing regimens. Likewise, in DIO mice, GLP-1 reference was evaluated at $C_{max}$ and $C_{min}$ following both acute and long-term treatment, and since no difference in glucose handling was observed between groups evaluated at $C_{max}$ and $C_{min}$, respectively, these groups were pooled and considered as one group in the respective figures and analyses. For the comparison of NN1177, NN1151 and NN1359 on glucose tolerance, vehicle-subtracted values were used to enable comparison across studies.

Statistical analysis was performed as one-way ANOVA followed by Dunnets post-hoc test. Statistics on body weight was performed as one-way ANOVA at the last day of treatment. Statistics on energy expenditure data was calculated using two-way ANOVA with repeated measures with Bonferroni's post hoc test. Statistical analysis was performed on average light-dark period values (only raw data, no mean data is presented). $p < 0.05$ was considered statistically significant. GraphPad Prism 6.4 (GraphPad) was used for all *in vivo* data visualization as well as statistical analyses.

## Results

### *In vitro* characterization

**Receptor binding.** NN1177 was confirmed to bind with high affinities to both glucagon and GLP-1-receptors from all species tested (Table 1). Despite minor differences in binding affinities, the balance between glucagon and GLP-1 receptor affinity was well preserved across assays with NN1177 showing two-to four-fold higher affinity to GLP-1 receptors than to glucagon receptors. In all species, NN1151 showed the highest glucagon receptor selectivity while NN1359 showed the highest GLP-1 receptor selectivity.

**Receptor potency.** Signalling through the GLP-1 and glucagon receptors was confirmed in cAMP-based *in vitro* functional assays (Table 2 and S1). In line with receptor binding results, the GLP-1 receptor preference of NN1177 was well conserved between species (1:3, 1:3 and 1:2 on human, rat and mouse receptor, respectively). On mouse receptors, NN1151, showed a glucagon receptor preference (1:0.4) while NN1359 showed a GLP-1 receptor preference (1:30). The *in vitro* potency assays were repeated in the presence of 100% species-specific plasma to assess the effects of protein binding on potencies (Table 2). Presence of plasma in the assay did not affect receptor balance (1:4, 1:0.2, 1:15, NN1177, NN1151, NN1359, respectively) but did cause a loss of potency. The reduced potency is likely explained by compound-albumin binding interfering with compound-receptor interaction as has been explained previously for similarly protracted peptides [23, 24]. In mouse plasma the apparent reduction for

**Table 1. Receptor binding.**

| | GLP-1 IC$_{50}$, nM (95% CI) | Glucagon IC$_{50}$, nM (95% CI) | NN1177 IC$_{50}$, nM (95% CI) | NN1151 IC$_{50}$, nM (95% CI) | NN1359 IC$_{50}$, nM (95% CI) |
|---|---|---|---|---|---|
| **Human GLP-1R** | 0.22 (0.13–0.28) | n/m | 0.64 (0.44–0.94) | 1.52 (0.77–2.95) | 0.84 (0.67–1.06) |
| **Human GCGR** | n/m | 0.30 (0.22–0.40) | 2.54 (1.54–4.19) | 0.81 (0.52–1.28) | 7.54 (3.96–14.36) |
| **Mouse GLP-1R** | 0.30 (0.27–0.33) | n/m | 0.11 (0.10–0.12) | 1.87 (1.46–2.41) | 0.37 (0.36–0.39) |
| **Mouse GCGR** | n/m | 0.60 (0.44–0.81) | 0.29 (0.27–0.32) | 0.11 0.11–0.12) | 14 (11.9–16.4) |
| **Rat GLP-1R** | 0.22 (0.20–0.25) | n/m | 0.17 (0.15–0.18) | 1.72 (1.47–2.01) | 2.06 (1.34–3.16) |
| **Rat GCGR** | n/m | 0.58 (0.56–0.60) | 0.30 (0.23–0.39) | 0.27 (0.25–0.30) | 6.29 (6.22–6.36) |

Receptor binding to human, mouse and rat GLP-1 and glucagon receptors (GCGR) performed in 1% species specific plasma; n/m, not measured.

**Table 2. Receptor potency.**

| Compound | GLP-1R EC$_{50}$, pM (95% CI) 0% plasma | GCGR EC$_{50}$, pM (95% CI) 0% plasma | Normalized Ratio 0% plasma[a] | GLP-1R EC$_{50}$, pM (95% CI) 100% plasma | GCGR EC$_{50}$, pM (95% CI) 100% plasma | Normalized Ratio 100% plasma[a] |
|---|---|---|---|---|---|---|
| | | | Potency | | | |
| | | | **Human receptors** | | | |
| **NN1177** | 0.92 (0.54–1.57) | 1.83 (1.35–2.48) | 1:3 | 149 (82–271) | 312 (137–710) | 1:3 |
| **GLP-1** | 1.86 (1.60–2.17) | n/a | n/a | n/a | n/a | n/a |
| **Glucagon** | 133 (116–152) | 1.46 (1.14–1.87) | n/a | n/a | n/a | n/a |
| | | | **Mouse receptors** | | | |
| **NN1151** | 1.13 (0.82–1.55) | 0.90 (0.47–1.71) | 1:0.4 | 161 (86–300) | 60.3 (46.1–79.0) | 1:0.2 |
| **NN1177** | 0.49 (0.29–0.80) | 1.96 (1.57–2.46) | 1:2 | 12.6 (8.2–19.4) | 103 (71–151) | 1:4 |
| **NN1359** | 0.51 (0.33–0.79) | 29.7 (21.7–40.5) | 1:30 | 95.3 (85.8–105.8) | 2540 (1820–3540) | 1:15 |
| **GLP-1 reference** | n/a | n/a | n/a | 24.6 (16.8–35.9) | 442,000 (239,000–818,000) | n/a |
| GLP-1 | 1.03 (0.94–1.13) | n/a | n/a | n/a | n/a | n/a |
| Glucagon | 92.7 (75.8–113.3) | 1.88 (1.61–2.21) | n/a | n/a | n/a | n/a |

Receptor potency at human and mouse GLP-1 and glucagon receptors (GCGR) in absence and presence of species-specific plasma; n/a, not applicable.

[a]Ratios were calculated on EC$_{50}$ levels after normalization to the endogenous ligand on each receptor, in absence of plasma, to account for slight differences in the assay systems and reported as the ratio of GLP-1R:GCGR EC$_{50}$ values.

NN1177 was significantly smaller than in human and rat plasma rendering a higher *in vitro* receptor potency in this species. The GLP-1 reference compound was verified to be a potent and selective GLP-1 receptor agonist at mouse and rat receptors (Table 2 and S1).

### *In vivo* characterization

**Body weight and body composition.** In DIO mice, 4–5 weeks of daily s.c. administration of the three co-agonists reduced body weight in a dose-dependent manner (Fig 2). Vehicle-treated mice remained weight stable while GLP-1 reference (5 nmol/kg) reduced body weight by 10%-17% (p<0.001) during the study period. As *in vitro* data indicated a very high mouse GLP-1 receptor potency, NN1177 was evaluated in a narrow dose interval (3–5 nmol/kg). Indeed, NN1177 was found to normalise body weight (22–33% body weight reduction) already at the lowest dose. The compound with highest glucagon receptor-selectivity (NN1151), induced a steep dose response and a continuous weight loss with no tendency to plateau. No effect was observed at the lowest dose NN1151 (1 nmol/kg) while at the two highest doses (10 and 30 nmol/kg) the majority of mice had to be terminated prematurely as the maximum ethically tolerated weight loss was reached. In contrast, a shallower dose response curve was observed for the GLP-1 receptor-preferring analogue, NN1359, where all doses evaluated (1–30 nmol/kg) induced significant body weight loss of 10–30% reaching a plateau during the last weeks of treatment (Fig 2). Reductions in body weight for all three co-agonists were mainly driven by loss of fat mass (58–76%, S2 Fig).

In DIO rats, three weeks of daily s.c. administration of NN1177, GLP-1 reference or combination of the two induced a weight loss of 20% (p<0.001 vs vehicle and GLP-1 reference), 5% (p<0.05 vs vehicle) and 30% (p<0.001 vs vehicle; p<0.01 vs NN1177 alone), respectively (Fig 3). In contrast to DIO mice, for both NN1177 and the combination of NN1177 and GLP-1 reference, a large part of the body weight loss in the DIO rats was driven by loss of lean mass as only 24–29% of the observed weight loss could be explained by loss of fat mass (S2 Fig).

**Food intake and energy expenditure.** In DIO mice, sub-chronic exposure of all co-agonists reduced food-intake in a dose-dependent manner. Significantly larger weight losses were

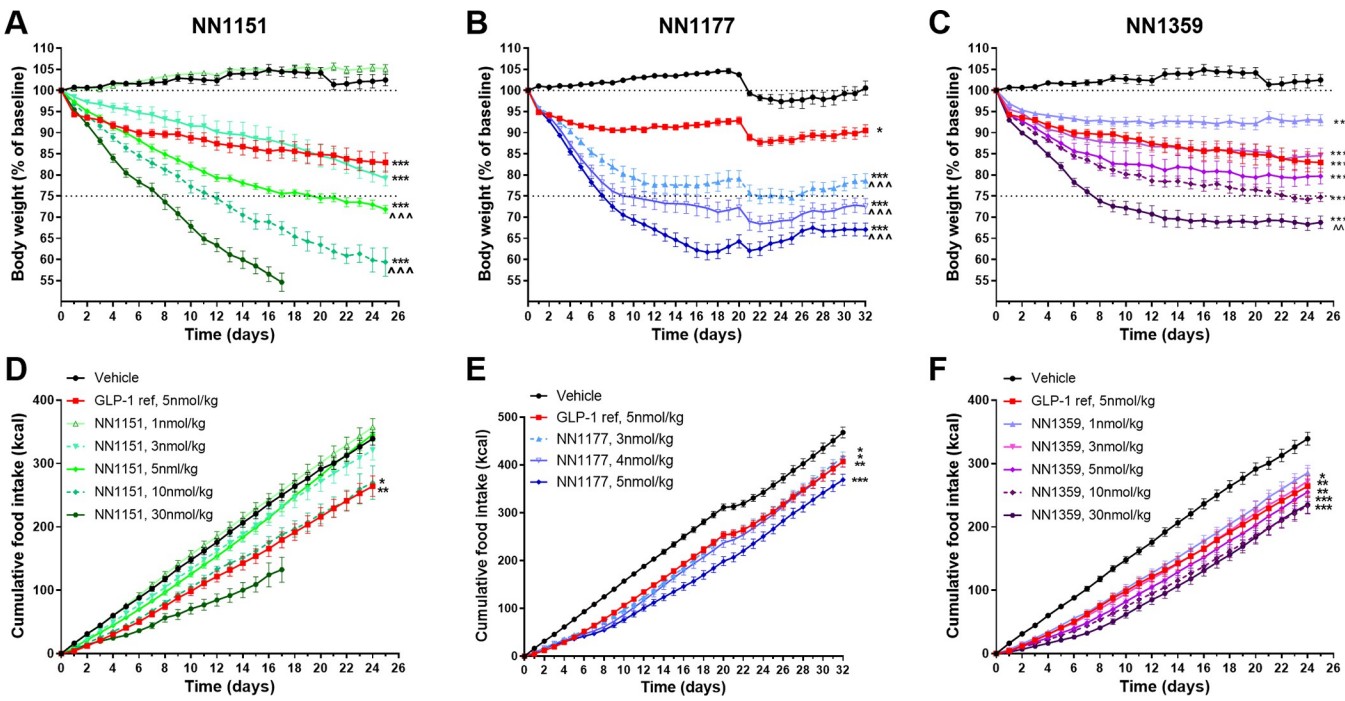

**Fig 2. Effect on body weight and food intake in DIO mice.** Body weight as percent of baseline in DIO mice with a baseline body weight of 45–50 g (A, B, C) and cumulative food intake in kcals per animal (D, E, F) of the GLP-1/glucagon receptor co-agonists NN1151 (A, D), NN1177 (B, E) and NN1359 (C, F) vs vehicle and the GLP-1 reference compound, following daily s.c. administration in DIO mice (n = 8–20) *p<0.05; **p<0.01; ***p<0.001 vs vehicle; ^^p<0.01, ^^^p<0.001 vs the GLP-1 reference compound as assessed on the last day of treatment. For clarification where hard to differentiate statistical denotations in the figure (D): p<0.05 NN1151 10nmol/kg vs vehicle; p<0.01 GLP-1 ref vs vehicle; (E): p<0.05 GLP-1 ref vs vehicle and NN1177 3nmol/kg vs vehicle; p<0.01 1177 4nmol/kg vs vehicle; p<0.001 NN1177 5nmol/kg vs vehicle; (F): p<0.05 NN1359 3nmol/kg vs vehicle; p<0.01 GLP-1 ref vs vehicle and NN1359 5nmol/kg vs vehicle; p<0.001 NN1359 10nmol/kg vs vehicle and NN1359 30nmol/kg vs vehicle.

observed for NN1177 and NN1151 (p<0.001 vs GLP-1 ref) than for NN1359 and the GLP-1 reference despite comparable or lower reduction in food intake (Fig 2).

In line with above-mentioned results, in DIO rats NN1177 and the GLP-1 reference showed equal reduction in cumulative food intake despite a significantly greater weight-loss induced by NN1177 (Fig 3). The combination of NN1177 and the GLP-1 reference further reduced both food intake and weight-loss (Fig 3).

Indirect calorimetry in DIO mice treated for nearly two weeks with NN1177 revealed an increased energy expenditure per body weight (p<0.01 day 3; p<0.001 day 5–11) during both the light and the dark period (S3 Fig). During the treatment period, mice lost approximately 30% of their body weight. While energy expenditure increased, the respiratory exchange ratio (RER) decreased significantly during both the light and the dark period (p<0.001), reflecting a shift in substrate utilization towards fat oxidation. Physical activity was not affected by treatment.

**Glucose tolerance.** In DIO mice, glucose tolerance was evaluated in the acute setting (a single injection of NN1177), and at steady state (3–5 weeks daily dosing). IPGTT was performed at a timepoint of high ($C_{max}$) and low ($C_{min}$) compound exposure (5h and 24h post-administration of compound, respectively).

In the acute setting, mice dosed with NN1177 showed improved basal blood glucose (p<0.001) and glucose tolerance (p<0.001) compared with vehicle-treated animals at $C_{max}$ and at $C_{min}$ (Fig 4). At the lower dose of NN1177 (3nmol/kg), glucose tolerance was significantly improved when evaluated at $C_{max}$ than at $C_{min}$ (p<0.001). For the higher dose (5 nmol/

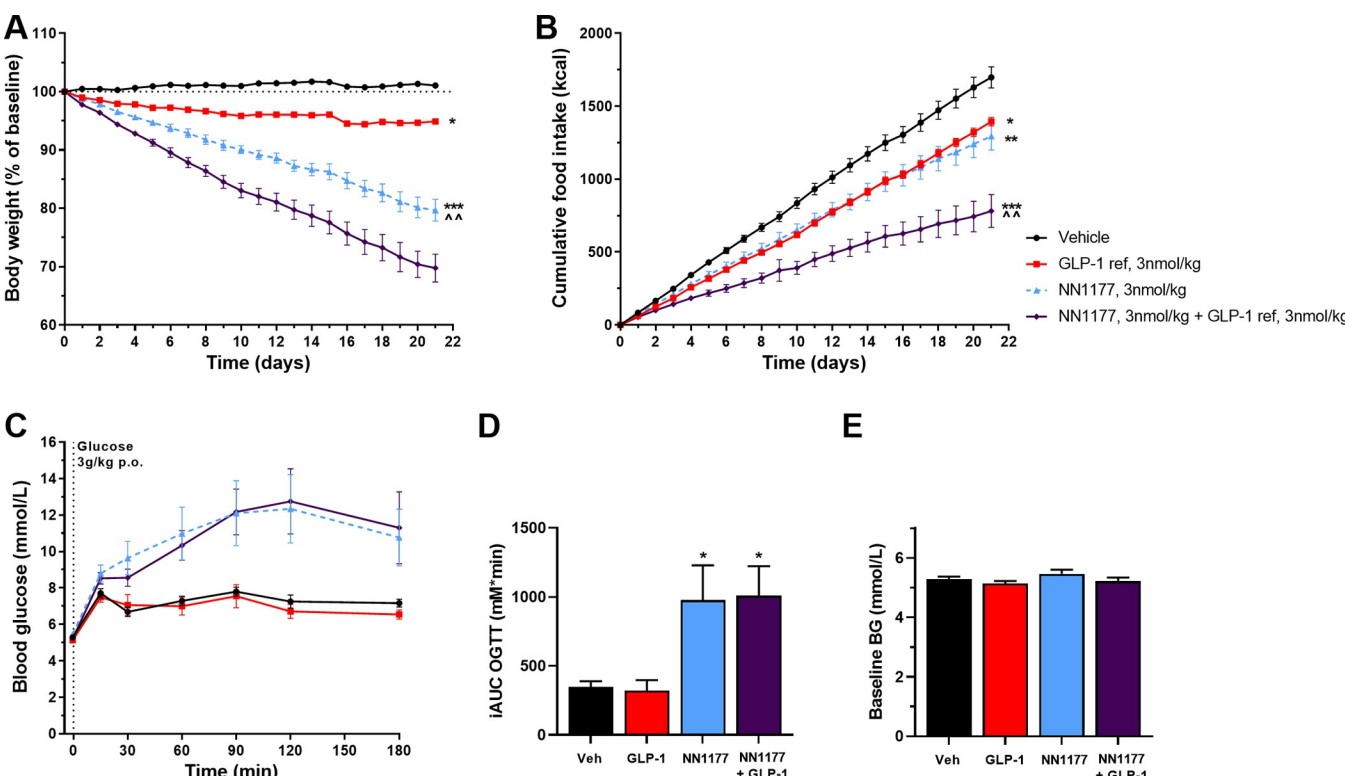

**Fig 3. Effect of NN1177 in DIO rats.** Body weight as % of baseline in DIO rats with an initial body weight of approximately 600 g (A), accumulated food intake in kcals (B), blood glucose levels during OGTT (C), incremental AUC of blood glucose levels during OGTT (D), and baseline blood glucose levels (E) in DIO rats treated once daily s.c. with vehicle, the GLP-1 reference, the GLP-1/glucagon receptor co-agonist NN1177 or a combination of NN1177 and the GLP-1 reference (n = 10). The oral glucose tolerance test was performed on day 15, 5 h post compound administration, corresponding to $C_{max}$.*p<0.05,**p<0.01, ***p<0.001 vs vehicle; ^^p<0.01 vs GLP-1 reference.

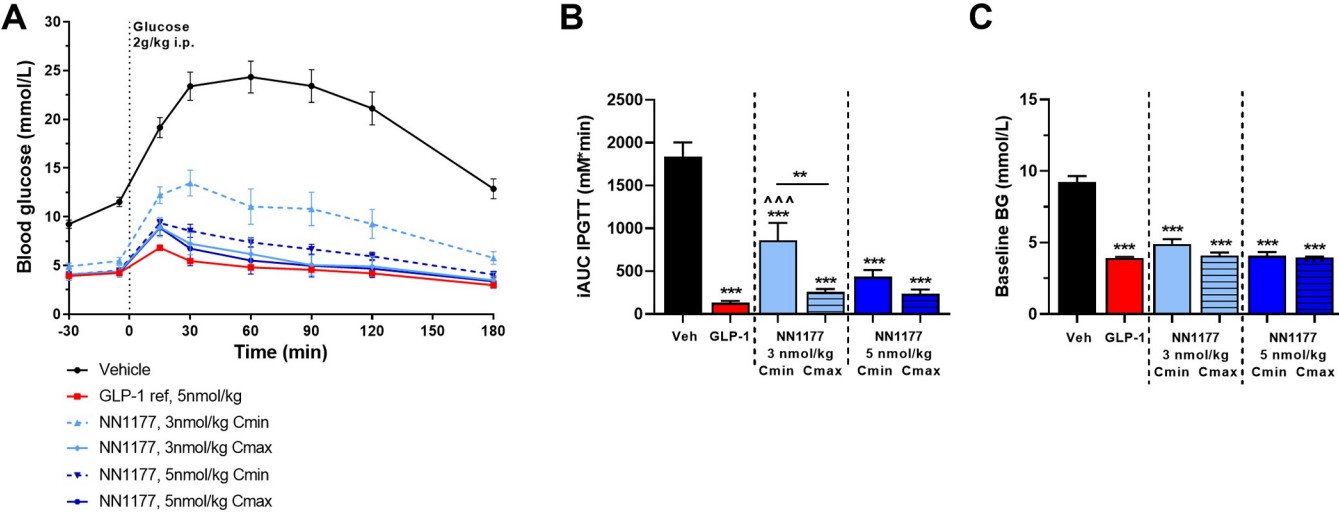

**Fig 4. Acute glucose control in DIO mice.** Blood glucose excursions (A) and incremental AUCs (B) following IPGTT in DIO mice receiving a single dose of vehicle, the GLP-1 reference compound or the GLP-1/glucagon receptor co-agonist NN1177 at 3 or 5 nmol/kg (n = 8–18). Glucose tolerance of NN1177 was evaluated 5 h post administration of compound ($C_{max}$) or 24 h post administration of compound ($C_{min}$). **p<0.01; ***p<0.001 vs vehicle or as indicated; ^^^p<0.001 vs the GLP-1 reference compound.

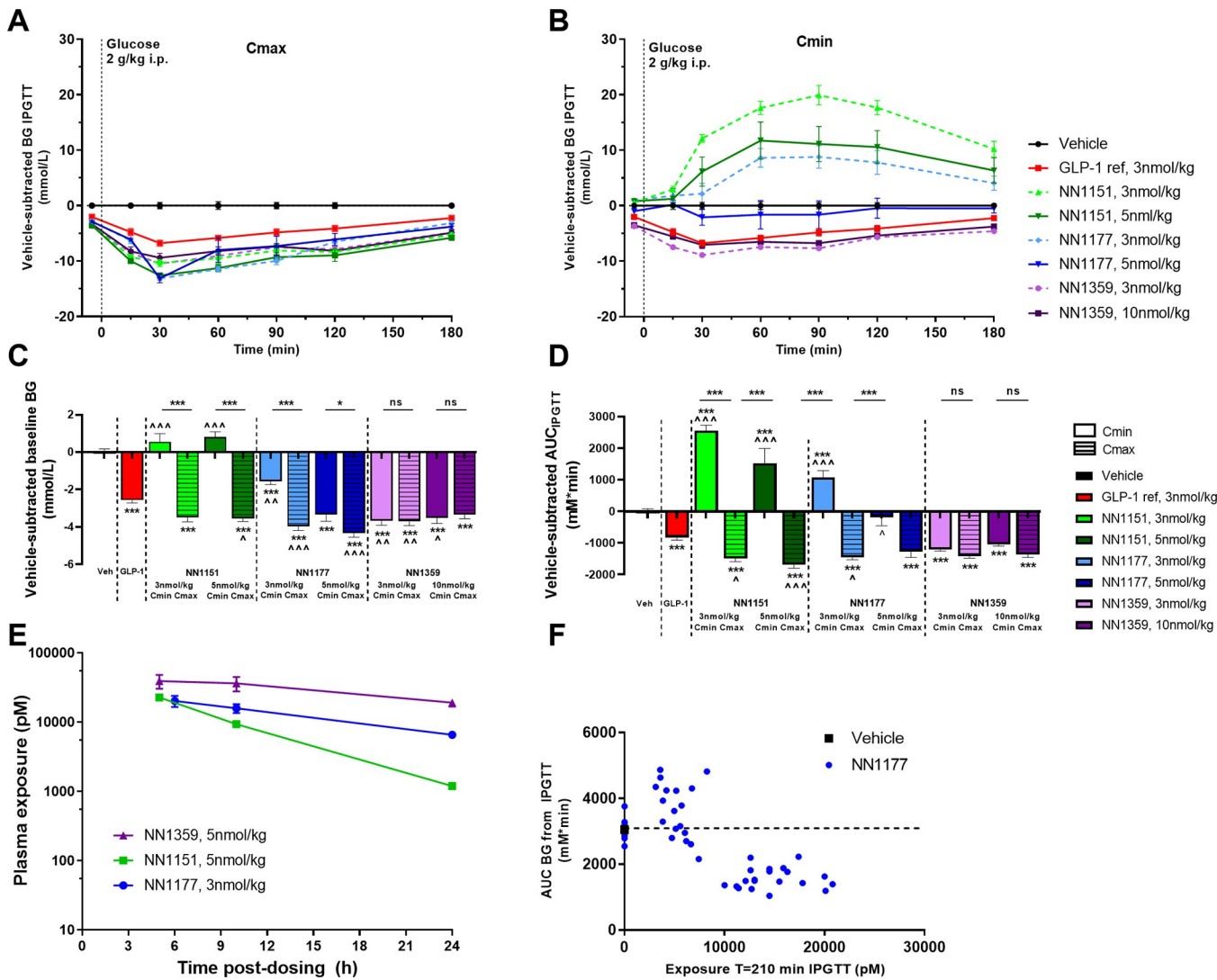

**Fig 5. Subchronic glucose control in DIO mice.** IPGTT in DIO mice treated sub-chronically with vehicle, the GLP-1 reference compound, the GLP-1 glucagon receptor co-agonists NN1151, NN1177 or NN1359. Vehicle-subtracted blood glucose excursions (A, B), corresponding fasted basal blood glucose (C) and derived AUCs of blood glucose during IPGTT (D). Exposure levels during steady state treatment (n = 4) (E) and exposure at t = 210 min vs AUC blood glucose (F). Glucose tolerance of the co-agonists was evaluated in steady state following 3–5 weeks of treatment, 5 h post administration of compound ($C_{max}$) or 24 h post administration of compound ($C_{min}$) respectively (n = 8–28). *p<0.05, ***p<0.001 vs vehicle or as indicated; ^p<0.05, ^^p<0.01, ^^^p<0.001 vs the GLP-1 reference compound.

kg), glucose tolerance was significantly improved at both $C_{max}$ and $C_{min}$, to the level of the GLP-1 reference.

In steady state, mice treated with both 3 and 5 nmol/kg NN1177 at $C_{max}$ showed improved basal blood glucose at (p<0.001 vs vehicle) and glucose tolerance (p<0.001 vs vehicle) comparable with the glucose control observed for the GLP-1 reference (Fig 5). At $C_{min}$, basal blood glucose was improved (p<0.001 vs vehicle) while mice treated with the lower dose of NN1177 (3 nmol/kg) showed impaired glucose tolerance compared with vehicle (p<0.001) and mice treated with the higher dose (5 nmol/kg) had glucose tolerance at the level of vehicle (Fig 5). The relation between NN1177 exposure at the time of the glucose tolerance test and glucose response is depicted in Fig 5F, showing that glucose response was impaired or comparable to

vehicle at exposure levels below approximately 10 nM and improved at exposure levels above approximately 10 nM.

Also, the co-agonists, NN1359 and NN1151, with higher GLP-1 receptor selectivity and higher glucagon receptor selectivity, respectively, were evaluated during steady state in a similar way as NN1177. For NN1359-treated mice basal glucose (Fig 5C) and glucose tolerance (Fig 5A, 5B and 5D) was significantly improved compared with vehicle (p<0.001) and comparable to the GLP-1 reference both at $C_{min}$ and $C_{max}$. For NN1151, basal blood glucose and glucose tolerance were improved at $C_{max}$ (p<0.001) (Fig 5A, 5C and 5D), while at $C_{min}$ basal blood glucose was at the level of vehicle treated animals and glucose tolerance was significantly impaired compared to vehicle (p<0.001), most pronounced at the low dose (Fig 5B–5D). 24-h exposure coverage, was confirmed for all the co-agonists in DIO mice in steady state (Fig 5E).

In DIO rats, oral glucose tolerance of NN1177 was evaluated at steady state (3 weeks daily dosing) at $C_{min}$ and $C_{max}$. While basal blood glucose was comparable between groups, NN1177 treatment significantly impaired glucose tolerance both when evaluated at $C_{max}$ and $C_{min}$ (p<0.05 vs vehicle) (Fig 3). Adding additional GLP-1 to NN1177 did not improve glucose tolerance, in spite of the greater weight loss achieved.

## Discussion

In the present study, we report the pre-clinical evaluation of NN1177, a GLP-1/glucagon receptor co-agonist. Evaluation was performed *in vitro* on human, mouse and rat GLP-1 and glucagon receptors, and in rodent models *in vivo*. A GLP-1 reference compound and two other co-agonists with different GLP-1-to-glucagon receptor ratios were evaluated in parallel in obese mice to further highlight the contribution from the respective receptors. All compounds evaluated were confirmed to be long-acting and suitable for once daily dosing in rodents as confirmed in pharmacokinetic studies (S2 File).

GLP-1-induced weight loss is known to be mediated mainly through reduction in food intake. In contrast, a major part of the weight-lowering effect of glucagon seems to be through increased energy expenditure, which has been described in both acute and long-term pre-clinical studies [25]. In line with these studies, we show that with more glucagon receptor preference less of the weight loss can be explained by reduction in food intake. In addition, a steeper weight loss with no tendency to plateau was observed with NN1151, the compound with highest glucagon receptor preference.

In humans, the weight loss induced by the glucagon component seem to translate well, as this is one of the most commonly presenting features in glucagonoma patients [26]. The effect of glucagon and oxyntomodulin administration on energy expenditure has however mainly been confirmed in acute studies while longer-term studies are scarce [13–15]. Furthermore, the underlying mechanism of glucagon-induced increase in energy expenditure is not yet clear and species differences have been reported [27, 28]. In particular, activation of brown adipose tissue (BAT) seems to translate poorly across species [29]. In rodents, glucagon increases BAT activity, but in humans the glucagon-induced increase in energy expenditure seems independent of BAT thermogenesis [30]. Also, mice and rats seem to respond differently to glucagon-induced weight loss. In the present study, NN1177 caused pronounced body weight loss in both mice and rats with a clear contribution of energy expenditure in both species. In rats most of the weight loss was caused by loss of lean mass whereas in mice primarily fat was lost. It has previously been reported that BAT cells isolated from rats are 200-fold more sensitive to glucagon compared with mouse BAT cells [31]. The higher grade of tissue wasting seen in rats could therefore be caused by a higher demand of thermogenic substrate in this species. Loss of lean-body mass is commonly also seen in patients with glucagonoma [12, 32].

Another difference between mice and rats was observed when evaluating glucose tolerance. In contrast to NN1177-treated mice, rats were found to have impaired glucose tolerance at all NN1177 doses tested and regardless of compound exposure at time of evaluation. Even rats receiving GLP-1 reference on top of NN1177 failed to improve glucose tolerance. A difference in NN1177 *in vitro* receptor balance for mouse and rat GLP-1 and glucagon receptors could hypothetically be the explanation, if NN1177 glucagon receptor preference was found to be higher for rat compared with mouse. However, for NN1177 there was no such difference found between mouse and rat receptor profiles. Thus, rats seem more sensitive to glucagon than mice. Species differences in receptor expression levels, and receptor tissue distribution could be part of the underlying explanation. When comparing the tissue distribution of glucagon receptors in rats and mice, it seems to be more widespread in rats, where glucagon receptor mRNA expression has been reported in the heart, spleen, stomach and thyroid of rats but not mice [33–36] (S2 Table). The difference in receptor distribution could possibly explain the different results obtained in these species, by mediating different down-stream effects and/or by apparently changing the glucagon-to-GLP-1 receptor balance. Clearly if the rat is used as a preclinical model for co-agonist evaluation, the identified optimal GLP-1-to-glucagon receptor balance would be very different from that identified in the mouse, and likely with only minimal glucagon receptor agonism in the molecule. Human gene expression data-bases [https://www.gtexportal.org/home/faq#citePortal, https://www.proteinatlas.org/about/licence, https://tabula-sapiens-portal.ds.czbiohub.org/home] show a relatively similar tissue distribution of glucagon receptors in humans as that reported for mice (S3 Table). The tissue distribution of GLP-1 receptors seems more similar across humans, rats and mice [https://www.gtexportal.org/home/faq#citePortal, https://www.proteinatlas.org/about/licence, https://tabula-sapiens-portal.ds.czbiohub.org/home, Tabula Muris (czbiohub.org) [33–37], (S3 and S4 Tables). In addition to the differences in species complicating the evaluation and selection of clinical candidates, our data point to other potential pitfalls that could lead to selection of sub-optimal candidates. In several publications, glucose tolerance has been evaluated shortly after compound administration [16, 20] or after long term treatment but without indication of timing of compound administration in relation to the glucose tolerance evaluation [4, 38, 39]. Our data emphasize the importance of longer-time compound exposure, the necessity for evaluating glucose tolerance rather than a simple basal blood glucose measure, and awareness of compound exposure at the time of evaluation to get the complete picture of the co-agonist effect on glucose control. As shown in the present study, when glucose tolerance was evaluated acutely after a single dose of NN1177 (3 or 5 nmol/kg), mice had significantly improved glucose control, when evaluated both at minimal ($C_{min}$) and maximal ($C_{max}$) compound exposure. In long-term NN1177-treated mice, the trend for a difference in glucose control depending on time of evaluation ($C_{min}$ vs $C_{max}$) was more pronounced. While at $C_{max}$ glucose control was significantly improved, there was an impairment at $C_{min}$, most pronounced at the lower dose. Thus, it might be difficult to catch the impact of glucagon in acute studies and at $C_{max}$, which may explain why co-agonists with high glucagon receptor preference previously been reported optimal, i.e. able to reduce body weight and improve blood glucose control in the preclinical setting [20]. Further support to this hypothesis was gained when evaluating glucose tolerance following long-term treatment of two other co-agonists (NN1151 and NN1359). These two co-agonists have a different GLP-1-to-glucagon ratio, where NN1151 has a clear glucagon receptor preference and NN1359 a clear GLP-1 receptor preference. Interestingly, at $C_{max}$ glucose tolerance was improved for both co-agonists to the level of the GLP-1 reference. However, at $C_{min}$, glucose tolerance remained improved for NN1359 but was significantly impaired with NN1151. The most pronounced impairment was observed at the lowest

dose. Basal glucose measurements were found much less sensitive in depicting these dynamics but showed similar tendencies as seen in the glucose tolerance tests.

The differences in glucose tolerance performed at $C_{max}$ and $C_{min}$, seen with both NN1177- and NN1151-treatment, might be explained by a difference in threshold exposure levels for activation of the GLP-1 and glucagon receptors. For NN1177, exposure levels below approximately 10 nM seemed to correlate with impaired glucose tolerance while exposure levels above seemed to improve glucose tolerance. Given that glucagon receptor agonism impairs glucose tolerance while GLP-1 receptor agonism improves it, it appears that the threshold exposure level for GLP-1 receptor activity is higher than that of the glucagon receptor. Thus, when exposure levels are above the threshold for GLP-1 it is possible to counteract the glucagon-induced impairment of glucose tolerance. At lower exposure values the glucagon effect seems more dominant. This would also explain the better outcome of glucose tolerance at high dose compared to low dose-treated animals, for both NN1177 and NN1151. In line with the present study, Day *et al.* found that single administration of a co-agonist initially improved glucose tolerance, but over time, as exposure levels decreased, glucose tolerance was impaired [20]. The underlying cause for the different threshold activation levels for GLP-1 and glucagon receptors and difference in sensitivity to glucagon may be attributed to a difference in receptor expression level, in combination with differential tissue distribution, as described above. It is also possible that the receptor expression is changed following long-term exposure and may be the underlying reason for discrepancies between acute and chronic studies. Agonist-mediated receptor endocytosis and receptor de-sensitization which is commonly described for G-protein-coupled receptors [40] could over time change the expression balance between the two receptors. It is thus possible that the selected co-agonist with initial optimal balance is found sub-optimal over time.

In summary, the evaluation of GLP-1/glucagon receptor co-agonists poses a number of challenges. For many biologies the exact balance in a combination may not be critical, but when combining GLP-1 and glucagon the correct ratio seems to be crucial to achieve optimal weight loss combined with optimal glucose control. The pharmacodynamic effects at a given ratio differ between species, at least partly due to differences in glucagon receptor expression, tissue distribution and down-stream effects. It is difficult to predict which species translates best to humans. In addition, our data indicate that the optimal receptor balance depends on exposure level and is changed during time of treatment and thus the molecule with an initial optimal profile may become sub-optimal over time. This finding implies the need for longer, more thorough clinical studies in order to evaluate the true effect of the compound. The aim to preclinically identify a successful clinical candidate, a co-agonist with an optimal receptor balance in humans has indeed a long list of challenges and pitfalls, which is emphasized by the long list of co-agonists tested and terminated in clinic.

## Supporting information

**S1 Fig. Chemical structures.** Structures of the A) GLP-1 reference compound B) co-agonist NN1151, C) co-agonist NN1177 and D) co-agonist NN1359.
(TIF)

**S2 Fig. Effect of NN1177 on body composition in DIO mice and rats.** Delta body weight (A, C) and delta body fat (B, D) of DIO mice (A, B) and DIO rats (C, D) treated daily s.c. for 3–5 weeks with vehicle, GLP-1 reference, NN1177 or a combination of NN1177 and GLP-1 reference (n = 8–20). Percentage indicates the amount of weight loss that can be explained by loss of fat mass. *p<0.05, **p<0.01, ***p<0.001 vs vehicle; ^p<0.05, ^^^p<0.001 vs the GLP-1

reference compound.
(TIF)

**S3 Fig. Effect of NN1177 on energy expenditure in DIO mice.** Energy expenditure per body weight (A), respiratory exchange ratio (RER) (B) and activity (C) in DIO mice treated once daily s.c. with vehicle or NN1177 (n = 8). **p<0.01; *** p<0.001 vs vehicle, two-way ANOVA based on mean light/dark values (average values shown).
(TIF)

**S1 Table. Receptor potency at rat GLP-1 and glucagon receptors.**
(DOCX)

**S2 Table. Tissue distribution of glucagon receptors in rat and mouse.**
(DOCX)

**S3 Table. Tissue distribution of glucagon and GLP-1 receptors in humans.**
(DOCX)

**S4 Table. Tissue distribution of GLP-1 receptors in rat and mouse.**
(DOCX)

**S1 File. Compound synthesis.**
(DOCX)

**S2 File. Pharmacokinetics in mice and rats.**
(DOCX)

## Acknowledgments

We thank all Novo Nordisk employees involved in evaluating the compounds. Especially we thank Natasha Barascuk Michaelsen for bioanalysis support, Lene Martini for work on *in vitro* assays and Steffen Reedtz-Runge for input on chemical design. Moreover, Hanne Jensen-Holm, Jens Henriksen, Jing Han, Juan Lv, Karin Hamburg Albrechtsen, Kirsten Holmberg, Lisbeth Eriksen, Merete Munk Dam, Nille Hammerum and Xiaoqiao Zhang are acknowledged for excellent technical assistance in performing the *in vitro* and *in vivo* assays and preparing the formulations.

## Author Contributions

**Conceptualization:** Lotte Simonsen, Jesper Lau, Thomas Kruse, Per Sauerberg, Charlotta Dornonville de la Cour.

**Formal analysis:** Lotte Simonsen, Tingqing Guo, Jim McGuire, Jacob Fuglsbjerg Jeppesen.

**Investigation:** Tingqing Guo, Jim McGuire, Kristoffer Niss.

**Methodology:** Lotte Simonsen, Tingqing Guo, Charlotta Dornonville de la Cour.

**Project administration:** Per Sauerberg.

**Resources:** Jesper Lau, Thomas Kruse.

**Supervision:** Lotte Simonsen, Charlotta Dornonville de la Cour.

**Visualization:** Lotte Simonsen, Jesper Lau, Jim McGuire, Kristoffer Niss, Charlotta Dornonville de la Cour.

**Writing – original draft:** Lotte Simonsen, Charlotta Dornonville de la Cour.

**Writing – review & editing:** Lotte Simonsen, Jesper Lau, Thomas Kruse, Jim McGuire, Jacob Fuglsbjerg Jeppesen, Kirsten Raun, Charlotta Dornonville de la Cour.

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
