## [Decision Letter · Decision Letter 0]

14 Feb 2022

PONE-D-22-02029Preclinical evaluation of a protracted GLP-1/glucagon receptor co-agonist: Translational difficulties and pitfallsPLOS ONE

Dear Dr. Dornonville de la Cour,

Thank you for submitting your manuscript to PLOS ONE. After careful consideration, we feel that it has merit but does not fully meet PLOS ONE’s publication criteria as it currently stands. Therefore, we invite you to submit a revised version of the manuscript that addresses the points raised during the review process.

We look forward to receiving your revised manuscript.

Kind regards,

Michael Bader

Academic Editor

PLOS ONE

Journal Requirements:

"All authors are or were full time employees at Novo Nordisk, and hold a minor share portion as part of their employment."

We note that you received funding from a commercial source: Novo Nordisk

Reviewers' comments:

Reviewer's Responses to Questions

**Comments to the Author**

1. Is the manuscript technically sound, and do the data support the conclusions?

Reviewer #1: Yes

Reviewer #2: Yes

2. Has the statistical analysis been performed appropriately and rigorously? 

Reviewer #1: Yes

Reviewer #2: Yes

3. Have the authors made all data underlying the findings in their manuscript fully available?

Reviewer #1: Yes

Reviewer #2: Yes

4. Is the manuscript presented in an intelligible fashion and written in standard English?

Reviewer #1: Yes

Reviewer #2: Yes

5. Review Comments to the Author

Reviewer #1: Simonsen L et al. studied GLP-1/glucagon receptor co-agonists which have different affinities to each receptors. Major purposes are characterizing the compound 1177 but for, comparison, compounds 1151 and 1359 were also analyzed. Studies are well designed and experiments were conducted carefully. Data presentation and interpretation are fair. Unfortunately, the data showed that such co-agonists have many obstacles to proceed to clinical application. Most interestingly, there is clear differences between data obtained in mice and rats. The data are useful for researchers in this field. This reviewer wants to request to provide with more information about antibodies used for measuring compound exposure experiments described in lines 187-197.

Reviewer #2: I am impressed the works done by Novo Nordisk. This is a very interesting research concerning the GLP-1R/GCGR dual agonist development. Indeed, compared with GLP-1R/GIPR dual agonist, the progress of development of GLP-1R/GCGR dual agonist remains slow. Even though there is already a report on 2009 concerning the effects of the first GLP-1R/GCGR dual agonist. Currently, MEDI0382 is one of the most successfully GLP-1R/GCGR dual agonist, I think if it is possible, the authors can compared the body weight reduction effects of MEDI0382 with NN1177. I noticed that the fatty acid albumin binder on NN1177 is different with semaglutide, more discussion should added concerning the reason for selected this kind of fatty acid as albumin binder on NN1177.

6. PLOS authors have the option to publish the peer review history of their article (what does this mean?). If published, this will include your full peer review and any attached files.

Reviewer #1: No

Reviewer #2: No

---

## [Author Response · Author response to Decision Letter 0]

18 Feb 2022

1. We have ensured that headings, figures, references and tables are correctly formatted and named.

2. Thank you for pointing to the discrepancy between ‘funding information’ and ‘financial disclosure´. There was no external funding source for these studies. All studies were supported by Novo Nordisk A/S. The information has been corrected in the funding information.

3. Thank you for pointing to the lack of sponsor information. Please use this amended competing interest information: 

All authors are or were full time employees at Novo Nordisk A/S and hold a minor share portion as part of their employment. Novo Nordisk A/S manufactures and markets pharmaceuticals related to diabetes, obesity and other chronic diseases and has intellectual property rights for several inventions related to these diseases. This does not alter our adherence to PLOS ONE policies on sharing data and materials.

Reviewer #1:

We acknowledge the reviewer’s wish for including more information on antibodies used for compound exposure analysis. We have now added more details to the manuscript (Page 9, line 192-197)

Reviewer #2:

We agree with the reviewer that it would be interesting to compare body weight profiles of NN1177 with MEDI0382, as well as with other co-agonists that have made it into clinic. We do however feel this comparison is hard to make and conclusions will be somewhat speculative in the absence of head-to head data in similar in vitro assay systems. Based on the body weight profiles we can speculate on the receptor balance in competitor molecules, but we do not think such comparisons/speculations will contribute much to the increased understanding of our findings and is therefore not within the scope of the present manuscript.

We thank the reviewer for pointing out the lack of information relating to the choice of fatty acid linker. We have now elaborated on the reasoning, which is added to the manuscript (Page 5-6, lines 106-109) and have also included and additional reference (nr 22, Schlein et al. 2017) describing the technique used for physical stability measurements.

---

## [Editor Report · Decision Letter 1]

21 Feb 2022

Preclinical evaluation of a protracted GLP-1/glucagon receptor co-agonist: Translational difficulties and pitfalls

PONE-D-22-02029R1

Dear Dr. Dornonville de la Cour,

We’re pleased to inform you that your manuscript has been judged scientifically suitable for publication and will be formally accepted for publication once it meets all outstanding technical requirements.

Kind regards,

Michael Bader

Academic Editor

PLOS ONE
---

## [Editor Report · Acceptance letter]

24 Feb 2022

PONE-D-22-02029R1 

Preclinical evaluation of a protracted GLP-1/glucagon receptor co-agonist: Translational difficulties and pitfalls 

Dear Dr. Dornonville de la Cour:

I'm pleased to inform you that your manuscript has been deemed suitable for publication in PLOS ONE. Congratulations! Your manuscript is now with our production department. 

Kind regards, 

on behalf of

Prof. Michael Bader 

Academic Editor

PLOS ONE